# Molecular Modelling Study and Antibacterial Evaluation of Diphenylmethane Derivatives as Potential FabI Inhibitors

**DOI:** 10.3390/molecules28073000

**Published:** 2023-03-28

**Authors:** Shaima Hasan, Kawthar Kayed, Rose Ghemrawi, Nezar Al Bataineh, Radwa E. Mahgoub, Rola Audeh, Raghad Aldulaymi, Noor Atatreh, Mohammad A. Ghattas

**Affiliations:** 1College of Pharmacy, Al Ain University, Abu Dhabi 64141, United Arab Emirates; 2AAU Health and Biomedical Research Center, Al Ain University, Abu Dhabi 64141, United Arab Emirates

**Keywords:** enoyl-acyl carrier protein transferase, FabI, diphenyl methane, triclosan, lead optimization, docking, molecular dynamic simulation, antimicrobial activity

## Abstract

The need for new antibiotics has become a major worldwide challenge as bacterial strains keep developing resistance to the existing drugs at an alarming rate. Enoyl-acyl carrier protein reductases (FabI) play a crucial role in lipids and fatty acid biosynthesis, which are essential for the integrity of the bacterial cell membrane. Our study aimed to discover small FabI inhibitors in continuation to our previously found hit MN02. The process was initially started by conducting a similarity search to the NCI ligand database using MN02 as a query. Accordingly, ten compounds were chosen for the computational assessment and antimicrobial testing. Most of the compounds showed an antibacterial activity against Gram-positive strains, while RK10 exhibited broad-spectrum activity against both Gram-positive and Gram-negative bacteria. All tested compounds were then docked into the saFabI active site followed by 100 ns MD simulations (Molecular Dynamics) and MM-GBSA (Molecular Mechanics with Generalised Born and Surface Area Solvation) calculations in order to understand their fitting and estimate their binding energies. Interestingly, and in line with the experimental data, RK10 was able to exhibit the best fitting with the target catalytic pocket. To sum up, RK10 is a small compound with leadlike characteristics that can indeed act as a promising candidate for the future development of broad-spectrum antibacterial agents.

## 1. Introduction

The frequent use of antibiotics caused bacterial genetic mutation and the development of resistance to these drugs. Each time a new antibiotic was introduced and used widely, a number of bacterial organisms deciphered how to resist the drug’s bactericidal effects by developing genome mutations or resistance genes [1,2]. This led to the production of a population of antibiotic-resistant organisms. The massive appearance of drug-resistant bacteria and the lack of other therapeutic alternatives terrify public health professionals and, most of the time, put them in a critical position when it comes to prescribing antibiotics [3]. Additionally, the consequent failure of antibiotic therapy, especially in intensive care units (ICUs), has led to hundreds of thousands of deaths annually [4]. Antibiotic resistant infection is believed to increase mortality by up to ten million deaths per year by 2050, with a total gross domestic product (GDP) loss of $100.2 trillion if proper precautions are not taken [5,6]. As a result, this resistance has led to a huge increase in the economic burden for both hospitals and patients, due to the prolonged hospital stays and higher costs required, and it has made antibiotic choices for infection control increasingly limited and more expensive [7]. Thus, the development of new antibiotics is crucial.

Fatty acid (FA) synthesis is an important process for the bacterial cell survival, especially because it is required for cell membrane synthesis [8]. As shown in Figure 1, the first step in the FA biosynthetic cycle is the condensation of malonyl-acyl carrier protein (ACP) with acetyl-CoA by FabH [9,10,11,12]. In subsequent rounds, malonyl-ACP is condensed with the growing-chain acyl-ACP (FabB and FabF). In the elongation cycle of the second step, FabG mediates ketoester reduction by NADPH-dependent β-ketoacyl-ACP reductase [13,14]. Subsequently, dehydration is carried by FabA or FabZ [15,16], which converts β-hydroxyacyl-ACP dehydrases to trans-2-enoyl-ACP. FabI is an enoyl-acyl carrier protein reductase (ENR) that catalyzes the conjugate reduction of an enoyl-ACP to the corresponding acyl-ACP using the cofactor NADPH or NADH as a hydride source [17,18,19]. Further rounds successively add two carbon atoms per cycle, finally leading to the synthesis of palmitoyl-ACP. Hence, the FabI catalysis is a rate-determining step for the overall biosynthetic pathway and therefore inhibiting such an enzyme could finally lead to the death of the bacterial cell [20].

Therefore, FabI enzymes introduce themselves as potential targets for the development of new antibacterial agents [21]. The significant variation in their structure and organization compared to the corresponding human enzyme makes them an attractive target [22,23]. Currently, there are no FabI inhibitors in the drug market and therefore the discovery of new FabI inhibitors would provide additional alternatives for clinicians and other health care providers while combating bacterial infections. In our previous work [24], we have recently discovered a small FabI inhibitor “MN02”, which is a small bisphenolic compound that belongs to the diphenylmethane family and exhibits antibacterial activity in the low micromolar range, with a broad spectrum of activity. This compound has interesting similarities with the standard FabI inhibitor triclosan [25], especially in that it was predicted to bind in a similar fashion, forming the hydrogen bonding and stacking interactions with the key tyrosine residue and the cofactor nicotinamide ring [26]. The fact that the diphenylmethane scaffold has never been explored as a FabI inhibitor makes MN02 act as a promising candidate for lead optimization, in order to develop more clinically useful antibacterial agents, which we performed in this work.

## 2. Results and Discussion

### 2.1. MN02 Derivatives

As part of our attempts to participate in the worldwide efforts combating the rapid spread of bacterial resistance, our team had identified MN02 as a promising lead FabI inhibitor and thus a potential antibacterial agent [24]. The lead optimization stage was initiated by searching the NCI (National Cancer Institute repository, Bethesda, MD, USA) ligand database [27] for new a set of derivatives or closely related structures, using a 2D search algorithm called the MACCS structural keys by molecular operating environment (MOE) software [28]. As shown in Figure 2, ten compounds with a similarity score of 85% and more were selected in order to assess their activity against multiple bacterial strains and then understand their mode of binding and inhibition through various molecular modelling techniques. All hits had been categorized based on their structures—mainly on the type of bisphenyl linker—into three different sets: the unsubstituted diphenylmethane set (RK02, RK04, RK10, MN02 and triclosan (TCL)), the substituted diphenylmethane set (RK06, RK07, RK08 and RK09) and the diverse set (RK01,RK03 and RK05).

### 2.2. Antimicrobial Assay

Based on the aforementioned similarity search, ten MN02 derivatives were obtained from the NCI ligand library and initially screened for their antimicrobial effect via the disk diffusion test. Interestingly, eight of the tested compounds (i.e., RK01-RK05 and RK08-RK10) were found to have an inhibitory effect against Gram-positive bacteria; *Staphylococcus aureus* (*S. aureus*), methicillin resistant *Staphylococcus aureus* (*MRSA*) and *Bacillus subtilis (B. subtilis*), and three of which (i.e., RK02, RK03 and RK10) also showed inhibitory activity against the Gram-negative bacteria *Escherichia coli (E. coli*) (as shown in Table 1).

Their minimum inhibitory concentration (MIC) values were identified through broth dilution method (listed in Table 2). The compound testing was conducted against *MRSA*, *B. subtilis*, *S. aureus* and *E. coli*. Interestingly, RK01–RK05 and RK08-RK10 exhibited an antibacterial activity against the challenging *MRSA* strain with MIC values ranging from 2 to 41.5 μg/mL, and against *S. aureus* with MIC values between 1.3 and 41.5 μg/mL. It was the same for the latter derivatives, except RK05, tested on *B. subtilis*, which gave MIC values ranging from 1.3 to 41.54 μg/mL. However, only RK10 showed a broader effect and was active against the Gram-negative strain (*E. coli*) with an MIC value of 47.64 μg/mL.

In comparison with our lead compound MN02 [24], RK09 and RK10 showed even greater potency against the Gram-positive strains (three-fold) with slightly lesser activity against *E. coli* (Table 2). Compared to Chloramphenicol and TCL, all hits showed better inhibition against the *MRSA* strain, but much lesser or no activity against *E. coli*. In conclusion, RK10 is our best hit since it exhibited a pronounced activity against not only Gram-positive but also Gram-negative bacteria. Furthermore, the obtained MIC values against *MRSA*, *B. subtilis* and *S. aureus* were low, ranging between 1.3 and 2.7 μg/mL (lower than those obtained with the lead MN02).

Our compounds showed better inhibition with Gram-positive bacteria compared with Gram-negative bacteria. This result was not surprising since it was shown that Gram-negative bacteria are more resistant than Gram-positive bacteria, and cause significant morbidity and mortality worldwide [29] This is due to their distinctive structure; Gram-positive bacteria have a thicker peptidoglycan layer in their cell wall compared to Gram-negative bacteria. This thicker layer makes it easier for certain compounds to penetrate and disrupt the bacterial cell membrane, ultimately inhibiting its growth or causing cell death. In contrast, the outer membrane of Gram-negative bacteria contains an additional layer of lipopolysaccharides that can act as a barrier, making it more difficult for some compounds to penetrate the cell membrane and reach their target. Therefore, the compounds that are effective against Gram-positive bacteria may not be as effective against Gram-negative bacteria [29].

The enzyme testing was performed for the most potent diphenylmethane derivatives of MN02 (i.e., RK04, RK09 and RK10) along with two other potent structural scaffolds RK03 and RK05, in order to confirm whether or not these compounds exert their activity through inhibiting the FabI enzyme. The enzymatic testing was carried out through an in-house validated assay using the *S. aureus* FabI enzyme (saFabI) and using MN02 as a positive control. As shown in Figure 3, compounds RK03, RK04, RK05, RK09 and RK10 are specific for FabI and act as FabI inhibitors. RK03, RK04 and RK09 showed complete inhibition for saFabI at 100 μM (same as MN02). Interestingly, this was in line with their antibacterial effect on *MRSA, B. subtilis* and *S. aureus*. RK05 exhibited no significant inhibition for the target enzyme, and this was correlated with the microbiological testing showing no bacterial growth inhibition. Interestingly, RK10 showed 59% inhibition of the saFabI enzyme, meaning that this compound’s high antibacterial activity (as shown in Table 2) is possibly caused via more than one mechanism. This finding is of particular interest to us as RK10 bears a high resemblance to the lead compound, with only the chloro substitutions moved from the para to meta position (Figure 2).

### 2.3. Assessment of Pharmacokinetic and Druglike Characteristics

The pharmacokinetic and druglike properties of our MN02 derivatives were evaluated using the SWISSADME software [30]. As shown in Appendix A (Appendix A), all our compounds were predicted to have a high GI absorption, as well as the ability to cross the blood–brain barrier (BBB). In contrast, they were found to be a P-gp substrate—a membrane protein that is highly expressed at BBB [31], which refluxes the compounds and prevents their accumulation in the brain, minimizing any unwanted side effects. The potential for drug–drug interaction was noticed with the MN02 derivatives as they all have shown a tendency to bind to at least two enzymes from the cytochrome P450 family. Our compounds have fulfilled many rules such as Lipinski’s [32], Veber’s [33], Ghose’s [34], Egan’s [35] and Mugge’s [36] for the druglike characteristics. Additionally, none of the derivatives were predicted as PAINS or reactive, apart from RK03 and RK05, which were shown to possess a polyaromatic scaffold and a Michael acceptor in their structures, respectively [37,38]. On the other hand, these small molecules seem to be small enough to act as leads, nonetheless, it is advisable to consider their hydrophobicity at the optimization stage, as they are already showing relatively high LogP values (>3.5). Overall, our compounds can be introduced as good candidates for the future development of saFabI inhibitors.

### 2.4. Glide Docking and MD Simulation

The molecular docking was carried out to determine the best poses of our hits in the saFabI binding pocket, as seen in Figure 4. The docking scores of the tested compounds are in the range of −6.03 kcal/mol to −9.2 kcal/mol (Table 3). To come up with more reliable and accurate conclusions, a 30 ns molecular dynamic (MD) simulation was conducted for the best poses generated by docking followed by the MM-GBSA scoring. Interestingly, after using this method, the computed binding energies of the MN02 derivatives appear to exhibit a better correlation with the experimental findings than the initial docking scores (Table 3). For instance, the compound RK07 obtained the best docking score among all tested compounds (−9.2 kcal/mol); however, it had the worst MM-GBSA score of −22.6 kcal/mol, which correlates well with its MIC value (i.e., no activity against saFabI bacterial strain). Interestingly, the same was noticed with RK02 and RK06, where high docking scores of −9.2 kcal/mol and −8.3 kcal/mol and low MM-GBSA scores of −26 kcal/mol and −25 kcal/mol were obtained, respectively, where the latter type of scoring has again shown a better correlation with their corresponding MIC values (Table 3). Interestingly, on the other hand, RK10 was able to score the highest MM-GBSA score (−32.9 kcal/mol), in line with its high antibacterial activity with an MIC value of 1.32 μg/mL, although its binding energy generated by docking was relatively low (−8.00 kcal/mol) when compared with its peers in the same family. Compared to the docking-related scoring functions, MM-GBSA has previously demonstrated a more significant correlation between the predicted scores and the actual inhibition constant (Ki) or IC50. [39]. Nevertheless, MM-GBSA lacks full accuracy when it comes to the closely related structures, as is the case with RK03, which showed an MIC value of 2.29 μg/mL against the *MRSA* strain (Table 2) and showed a low MM-GBSA score (23.47 kcal/mol). This raises the possibility that diphenylmethane derivatives may exhibit other antibacterial mechanisms.

Speaking about the structure-activity relationship, the high activity and high binding affinity of RK10 are possibly due to the presence of chloro-substitutions at the meta position comparing with the para position in MN02. Most likely then, the halogen substitution at the para position of the benzene ring provides a better fitting for the molecule at the binding site. On the other hand, while the halogen substitution aided the antibacterial activity, the ligand binding appears to be relatively restrained and weakened by the presence of a substitution on the diphenyl linker as compared to the unsubstituted ligand as in RK10. This reduction is most clearly seen in the carbonyl-linker compounds (i.e., RK06, RK07 and, to a lesser extent, RK08), when compared to methyl-substituted derivative RK09 [26,40,41].

What made this worse for RK06 and RK07 is the absence of any hydroxyl group at the ortho position, which, along with the carbonyl linker’s effect, seem to be chiefly responsible for diminishing the antimicrobial activity of these two compounds by pushing them away from the center of the active site when compared to the co-crystallized ligand triclosan, as shown in Figure 4 [42]. Additionally, having two hydroxyl groups on the aromatic rings as seen in RK09 and RK10 seems to boost the antibacterial activity when compared to the monohydroxylated compounds (RK06 and RK07), in contrast to what was previously shown by the closely well-known diphenyl ether derivative (triclosan), indicating for a slightly altered structure activity relationship (SAR) and hence for a slightly different binding (Figure 4). All in all, mono or dihydroxyl substitutions are important for making the necessary interactions with the enzyme binary complex, and having them at the ortho position along with no substitution on the methylene linker seems to perfect the binding, and consequently, the overall antimicrobial activity.

### 2.5. MD Analysis

MD simulation was extended further to 100 ns simulation, in order to gain additional insights into the overall stability, flexibility and compactness of the ternary complex. First, the root mean square of deviation values (RMSD) were calculated for the protein backbone and the ligand atoms to monitor the protein secondary structure stability and the ligand binding inside the FabI binding pocket [43]. Values less than 2 Å for the ligand and less than 3 Å for the protein backbone are deemed favorable, whereas higher values may indicate for certain inconsistencies in the simulated system or, simply, for an inconvenient ligand binding [44]. On the other hand, the root mean square of fluctuation (RMSF) depicts each protein residue’s movement and the fluctuation during the entire length of the MD simulation. High RMSF values indicate that residues fluctuate significantly during the MD simulation [45]. The radius of gyration (Rg) value indicates how structural changes affect the protein compactness after binding with the ligand [46]. High Rg values may indicate for certain unfolding happening in the protein tertiary structure.

#### 2.5.1. Unsubstituted Diphenylmethane

Figure 5 shows the RMSD, RMSF and RG plots for the unsubstituted set of ligands. The RMSD plot of the protein backbone was calculated (Figure 5a), where most of the systems appear to converge by 40 ns of the simulation, all of which showing acceptable RMSD values of less than 3 Å, with an exception for RK02, which goes slightly beyond that limit. In contrast, the bound ligand molecules seem to converge more rapidly while showing low RMSD values that do not exceed 2 Å on average, as shown in Figure 5b. Most notably, RK10 deviated the least from its initial configuration, which is evident by its very low RMSD values that did not exceed 0.5 Å on average, correlating very well with its high binding affinity predicted by MM-GBSA previously.

The complex structures were compared with their Apo structure, as seen in the RMSF plot, to see if there were any changes to the protein once it was bound with a ligand. Fluctuations for all structures are similar with a few distinctions denoted by black boxes (Figure 5c). The residues at region 1 (Ala 153 to Val 159) and region 2 (Pro 216 to Asp 222) showed lower RMSF values compared to the FabI Apo structure. Whereas the latter region is just a distant loop that is less likely to influence the enzyme stability and the catalytic activity, region 1 has the catalytic Tyr 157, which is a known key residue for the enzyme catalytic activity [47]. Both deviations, nonetheless, depict that the target enzyme is becoming more rigid upon ligand binding, when compared to the Apo structure. Furthermore, the Rg values for the complex structures and the Apo structure did not show any significance, indicating that the protein compactness did not feature any significant changes throughout the MD simulation (Figure 5d). All in all, these data clearly demonstrate the stability of these complexes throughout the MD simulation, most remarkably, the high correlation shown by our best hit (RK10) that was predicted via molecular modelling as a strong FabI inhibitor, in line with its experimentally proven antibacterial activity.

#### 2.5.2. Substituted Diphenylmethane Set

The RMSD values for the RK06, RK07, RK08 and RK09 ternary complexes were plotted and are shown in Figure 6a. The backbone amide of the RK06 and RK09 complexes demonstrated a steady increase in the RMSD values for the first 40 ns simulation. Both ligands finally reached the steady-state with RMSD values higher than 3Å, indicating a less stable system when compared to that in the RK07 and RK08 complexes, which showed an average RMSD value of 2.5 Å. The ligand’s RMSD showed RK08 was the most stable ligand among this set of compounds, while RK06 showed the maximum deviation than the initially docked conformation (Figure 6b), in line with the MM-GBSA and experimental data that showed RK08 was one of the most potent of this group and RK06 as an inactive compound.

Additionally, the RMSF of RK06 showed a high variation compared to the Apo structure and RK08, specifically at region 3 (Gly 104 to Ser 110) and region 4 (Gly 192 to Gly 203), where the latter was defined as the substrate binding loop [23,48]. RK08 showed a low RMSF value at region 1 (Figure 6c), which confined the ligand interaction with catalytic Tyr 157. RK09 and RK07 showed a quite similar fluctuation pattern to the Apo structure with one difference at region 2. Similar to the unsubstituted set’s Rg plot, all systems exhibited almost the same protein compactness (Figure 6d).

#### 2.5.3. Diverse Set

The complex RMSD of RK03 illustrated an unstable system for the first 60 ns, in contrast RK01 and RK05 reached the stable state after 10 ns from initiating the simulation (Figure 7a). RK01 showed a deviation at 90 ns where it reached 3 Å. As illustrated in Figure 7b, all the ligands were quite similar in the saFabI binding pocket, with some fluctuations that can be seen in RK03 after 20 ns. Quite typical RMSF plots are seen for the diverse set structures along with the Apo structure, except at the loop segment of FabI (Pro 216 to Asp 222), where RK05 has a low RMSF compared to other structures (Figure 7c). All systems have showed a good protein compactness (Figure 7d).

### 2.6. Pairwise Energy Decomposition Analysis

As we concluded from the previous analysis, among all MN02 derivatives, RK10 appears to have the best-performing ligand in terms of binding stability, affinity and inhibitory activity. One more analysis step was run to better understand the residues and NADPH interactions with RK10. Pairwise energy decomposition measures the interaction energy between the pairs of residues and ligands in the system. The analysis was applied for the key residues in the binding pocket, NADPH and RK10. Figure 8 illustrates the energy decomposition for the key residues and NADPH with RK10; interestingly, NADPH showed the highest interaction contribution with RK10, specifically electrostatic and vdw interactions. The key residue Tyr 157 also showed a significant interaction contribution with the ligand; this finding complies with the commercially available antibacterial triclosan, which has been reported to have an inhibitory activity against FabI enzymes [49]. The diphenyl ether triclosan occupies a position near NAD+; this aids in making π–π interaction with NAD+ also as being in close contact with the key residue Tyr 157 [18,50]. A good contribution was seen with Met 99, Leu 102, Try 147 and Ser 197.

Figure 9a and b illustrates the fitting of RK10 along with the cofactor NADPH inside the saFabI active site. The ligand interaction was obtained for RK10, which shows the interaction with Met 99, Tyr 157 and NADPH, the most contributed residue as found earlier (Figure 8c).

## 3. Method

### 3.1. Similarity Search

Our previously discovered compound MN02 was used as a query for the NCI ligand library similarity search. The MACCS structural key feature of MOE software was used as a similarity search engine, where the overlapping percentage was set to 85% [51]. Ten compounds were found to have a similar structure to MN02. These new derivatives were tested against a variety of bacterial strains for their antimicrobial activity and then were proceeded for the ligand docking into the saFabI binding site.

### 3.2. Antimicrobial Testing

#### 3.2.1. Disk Diffusion

Screening was initially conducted by the disk diffusion test against a selected bacterial strains (Gram-positive *S. aureus*, *MRSA* and *B. subtilis* and Gram-negative bacteria *E. coli*, as per the Clinical Laboratory Standards Institute and previous studies [52,53]. Sterile filter paper disks (6 mm, Whatman Number 1) were soaked in the test compounds dissolved in dimethyl sulfoxide (20 mM) and left to dry at room temperature in order to remove any residual solvent. A sterile cotton swab was dipped into the inoculum suspension (adjusted to 0.5 McFarland standard turbidity). Disks were then placed carefully on nutrient agar plates that had been already inoculated with micro-organisms. The plates were incubated at 37 °C for 18–24 h. The diameter of the visible zone of inhibition was measured to the nearest whole millimeter. Chloramphenicol (Sigma Aldrich) was used as a reference antibacterial agent and positive control.

#### 3.2.2. Minimum Inhibitory Concentrations

MICs for the MN02 derivatives were determined by using the broth dilution method as per the Clinical Laboratory Standards Institute [54]. The compounds were tested against the following Gram-positive micro-organisms: *S. aureus*, *MRSA* and *B. subtilis*; and against the Gram-negative bacteria, *E. coli*. The stock solutions were prepared in dimethyl sulfoxide. They were then added to the culture medium (nutrient broth; LABM, Lancashire, UK), to obtain a concentration range from 0.19 to 181.47 μg/mL. Subsequently, 1 × 106 CFU/mL (colony forming unit/mL) of micro-organisms were added separately to each test-tube, which were incubated at 37 °C for 18–24 h. A mixture of media with 1% *v/v* dimethyl sulfoxide was used as a negative control. Chloramphenicol was used as a reference antibacterial agent and positive control. The MIC values were determined as the lowest concentrations of compound, with no visible bacterial growth. All the measurements were carried out in duplicate.

### 3.3. Enzyme Inhibition Assay

The FabI enzyme was purchased from a commercial source (Mybiosource, San Diego, CA, USA) along with all needed material (substrate and cofactor) and subsequently, the enzyme inhibition assay was validated in our lab. In this assay, the enzymatic activity of FabI was measured as the reduction of NADPH and monitored by the change in absorbance at 340 nm. Assays were performed in 96- half-area plates in a final assay volume of 100 μL. The reaction mixture consisted of 100 mM sodium phosphate (pH 7.4), 0.25 mM crotonoyl-CoA (the substrate), 0.4 mM NADPH and 9.6 μg of *S. aureus* FabI. The reaction was initiated by adding the enzyme, then the absorption at 340 nm was measured after 10 min at room temperature.

### 3.4. Protein Preparation and Grid Generation

The co-crystallized structure of *S. aureus* FabI with NADPH and an inhibitor was obtained from the protein data bank (PDB, ID: 4FS3) [55]. All the water molecules were removed from the co-crystallized structure. The enzyme was prepared using the protein preparation wizard by MOE software [51]. Examining the protein for any missing atoms, residues and loops was followed by any necessary corrections. Further adjustments for the protein were carried out by the Maestro protein preparation module to set up the partial charges for all atoms and protonation states for any concerned ionizable groups. A grid block was generated using the receptor grid generation in Glide. The co-crystalized ligand at the active site of saFabI was considered as a centroid for the grid generation. The grid-generation process helped in providing an accurate binding score with efficient and fast ligand docking calculations [56].

### 3.5. Ligand Preparation and Molecular Docking of MN02 Derivatives

Before ligand docking into the saFabI active site, all ligands along with triclosan as a positive control were prepared using the LigPrep module in Maestro in order to generate all possible tautomers, ring conformations and stereoisomers, and then to energy minimize their 3D structures using the OPLS3 force field [57,58].

The prepared database was then docked into the previously identified binding site of the prepared saFabI structure. The extra precision (XP) algorithm [59] of the glide docking tools in Maestro was employed for conformational sampling. The Docking scores were then calculated via the Glide XP scoring function that includes terms for van der Waals, the hydrogen bond, electrostatic interactions, the desolvation penalty and the penalty for intra-ligand contact.

### 3.6. Molecular Dynamic Simulation and MM-GBSA Scoring

The top-ranked docked pose for each compound was further investigated via MD simulations, where ligand stability and binding within the active site was assessed and analyzed. The MD simulation helped in predicting more accurately the molecular interactions between the protein and ligand and examined the ligand’s thermodynamics and kinetics [60] over a certain period of time, as well as being a complementary experiment alongside biological tests. The crystal structure of saFabI in a complex of the cofactor NADPH and the co-crystallized ligand was obtained from the protein data bank (PDB ID: 4FS3) for building the initial solvated system. The protein was prepared and cleaned via the pdb4amber program, where all water molecules were removed using the ff19SB force field. The docked pose of the concerned ligand along with NADPH were prepared via the Antechamber program [61] using the Generalized Amber Force Field (GAFF) [62]. A system of the protein–co-factor–ligand complex was built using the Xleap program; the charge of the system was neutralized by adding Na+ counter ions and then immersed in a truncated octahedral box of TIP3P water with a distance of 14 Å away from the ternary complex.

The system was energy minimized via the pmemd program in the AMBER 18 software package [63], while the solute atoms were restrained with a force constant of 500 kcal mol^−1^ Å^−2^. Then the entire system was minimized without restrains for 1000 cycles. For the molecular dynamics simulation, the energy-minimized system was then heated to the desired temperature of 300 K under NVT condition, with a 10 kcal mol^−1^ Å^−2^ restraint on ligand atoms over 20 ps. Using the Langevin thermostat, the SHAKE algorithm was employed to all bonds including hydrogen atoms with a collision frequency of 1.0 ps^−1^. Finally, a production MD simulation run of 100 ns for each of the selected compounds was performed under NPT conditions with a target temperature of 300 K and pressure of 1 atm [64,65,66]. Coordinates were recorded every 2 ps throughout the trajectory. The MM-GBSA score was calculated to gain more insight to the binding affinity of Ligand_NADPH_protein complexes via considering the first 30 ns of the MD simulation.

### 3.7. ADME Screening

The SwissADME online server [67] was used to assess the compounds’ absorption, distribution, metabolism and excretion (ADME), as well as their leadlike and druglike properties; for instance, to predict the compounds’ gastrointestinal absorption, blood–brain barrier permeability, CYP450 inhibition, Lipinski’s rule of five and Veber’s rule, PAINS, reactivity, etc. [30].

### 3.8. MD Simulation Analysis

After completing the MD simulation, several analyses were performed on the obtained trajectories to examine the system stability and flexibility by calculating the root mean RMSD and RMSF. The Rg analysis was also calculated as another useful tool to ensure the system compactness and protein integrity. Additionally, a pairwise energy decomposition analysis was calculated for the best-performed derivative to gain an insight on the residues’ contribution to binding interaction throughout the 100 ns simulation.

## 4. Conclusions

Based on the structure of the previously discovered inhibitor of enoyl-acyl carrier protein reductases “MN02,” a 2D similarity search was performed against the NCI ligand library. As a result, ten compounds were carefully selected for microbiological testing. Eight of these were able to show an antibacterial activity against most of the tested Gram-positive bacteria (i.e., *MRSA*, *B. subtilis*, *S. aureus*). Most notably, RK10 demonstrated a broad spectrum of activity, inhibiting both Gram-positive and Gram-negative bacteria in the low to medium micromolar range. The binding modes of the compounds were investigated using docking and molecular dynamic simulation, while MM-GBSA calculations were used to estimate their binding energies. Interestingly, the best score (−32.9 kcal/mol) was attained by RK10, which is a diphenylmethane compound that is closely related in structure to the known FabI inhibitor triclosan. Studying the structure activity relationship of RK10, it seems that the monohydroxy or dihydroxy substitution at the ortho position is necessary for activity, halogen substitution on the para position increases the activity, while a substitution on the diphenyl linker decreases the activity. RMSD, RMSF, the radius of gyration and hydrogen bond analysis were also calculated for all the compounds. The results showed that RK10 had the best fitting and the highest affinity towards the saFabI enzyme, making the necessary key interactions with the cofactor and the key catalytic residue Tyr 157. In summary, the results of this work demonstrate RK10’s potential to act as a lead compound for clinically useful antibacterial agents.

## 5. Limitations

In order to prove that the observed antibacterial activity of the RKs compounds is due to their binding to FabI enzyme, assessing the inhibitory effect of all compounds on FabI was important. However, only RKs 3, 4, 5, 9 and 10 were tested.Only four bacterial species were tested (*MRSA*, *E. coli*, *B. subtilis* and *S. aureus*), three Gram positive and one Gram negative. Even though testing only four strains with only one Gram-negative bacteria was published in previous studies assessing antibacterial effects of drugs/compounds [68], a more extensive study of the antibacterial activities of the selected compounds would be needed to draw reliable conclusions on their potential use as antibiotics.

## Figures and Tables

**Figure 1 molecules-28-03000-f001:**
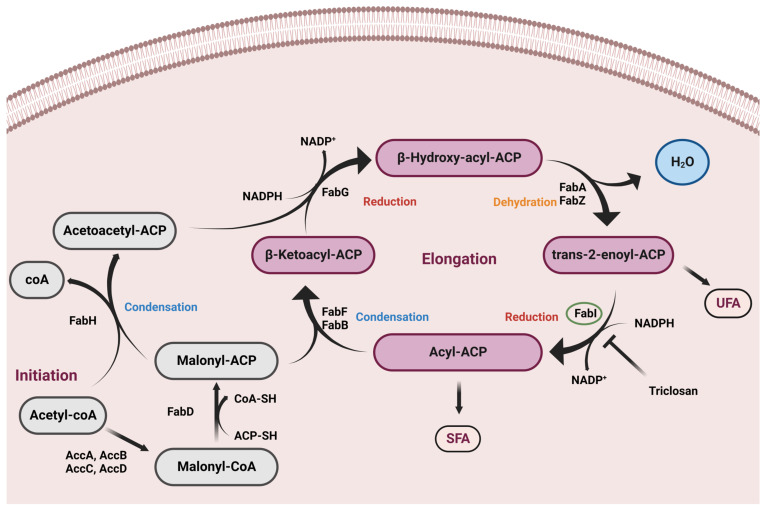
The bacterial biosynthetic cycle of fatty acids. The enzymes malonyl-CoA:ACP transacylase (FabD) and 3-oxooacyl-ACP synthase III (FabH) are involved in initiation of fatty acid synthesis, where 3-oxooacyl-ACP reductase (FabG), enoyl-ACP synthases (FabA and FabZ), enoyl-ACP reductase (FabI), and 3-oxoacyl-ACP synthases (FabB and FabF) are involved in fatty acid elongation. FabI (donated with a green circle) controls the rate of fatty acid synthesis. The inhibition of FabI enzyme will lead to fatty acid synthesis termination.

**Figure 2 molecules-28-03000-f002:**
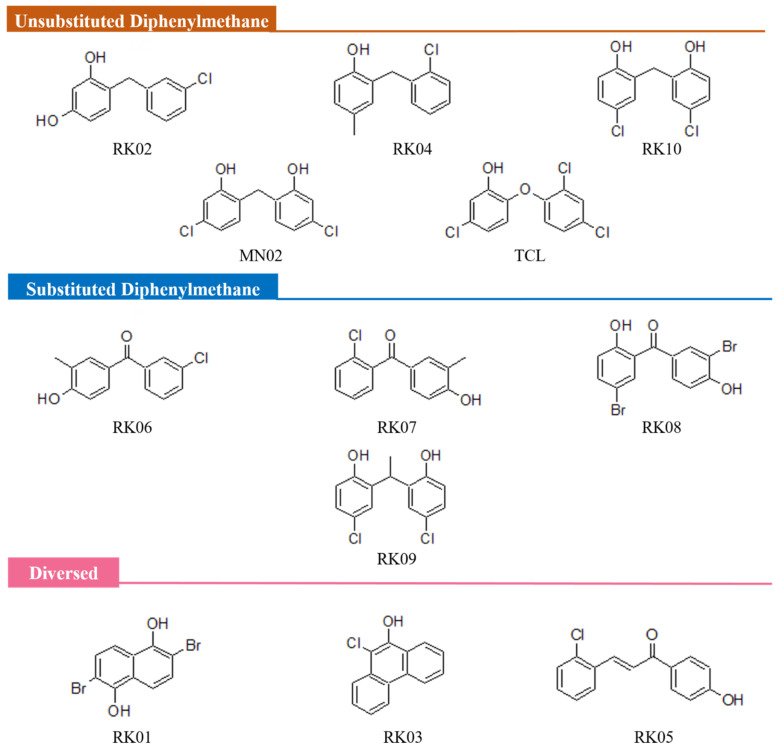
Structures of MN02, MN02 derivatives and triclosan. Based on their structures, the derivatives have been categorized into three sets: unsubstituted diphenylmethane, substituted Diphenylmethane and diverse.

**Figure 3 molecules-28-03000-f003:**
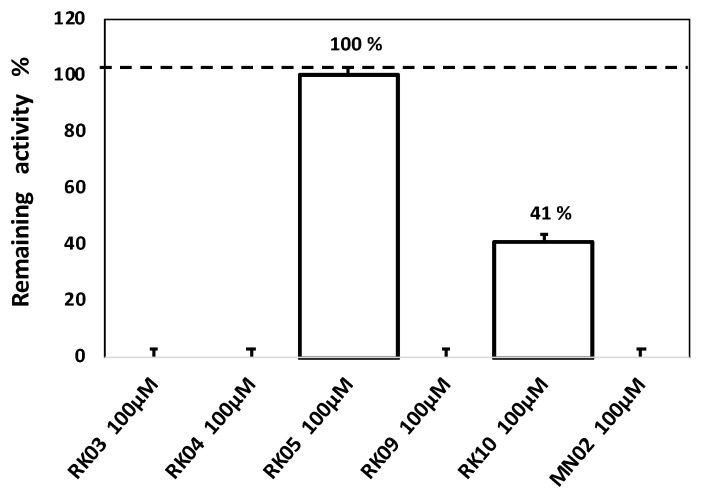
Inhibition activity of the tested compounds shown as remaining activity expressed in percentage of RK03, RK04, RK05, RK09, RK10 and MN02 as control.

**Figure 4 molecules-28-03000-f004:**
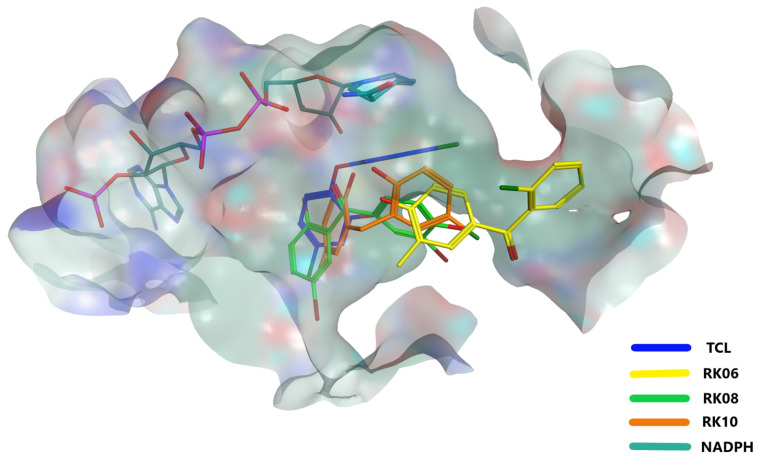
The predicted binding modes of RK06 (yellow sticks), RK08 (green sticks) and RK10 (orange sticks) along with the co-crystallized ligand triclosan (blue sticks) inside the FabI active site. RK10 and RK08 showed better fitting than RK06 due to the presence of the hydroxyl group at the ortho position.

**Figure 5 molecules-28-03000-f005:**
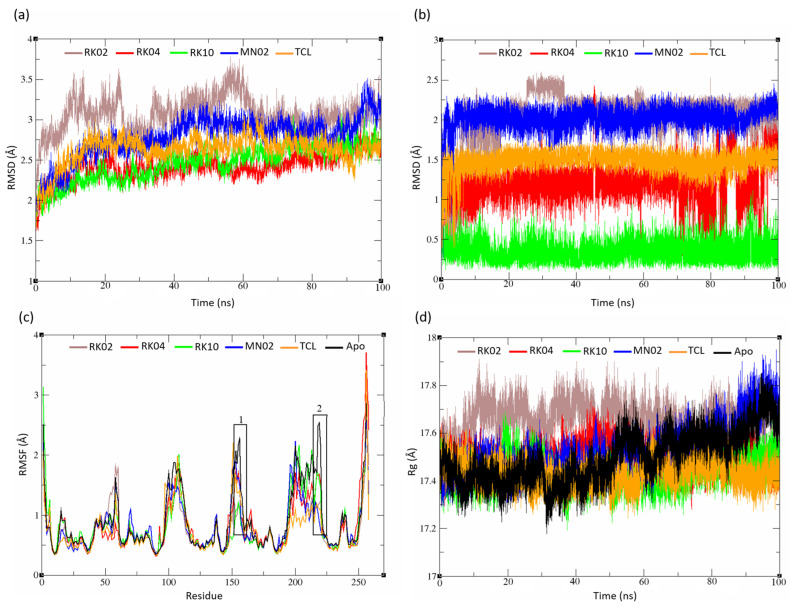
Unsubstituted Diphenylmethane set analysis. (**a**) Complex RMSD, (**b**) ligand RMSD, (**c**) backbone RMSF and (**d**) radius of gyration. The obtained data have shown that all complexes are stable throughout the 100 ns of MD simulation.

**Figure 6 molecules-28-03000-f006:**
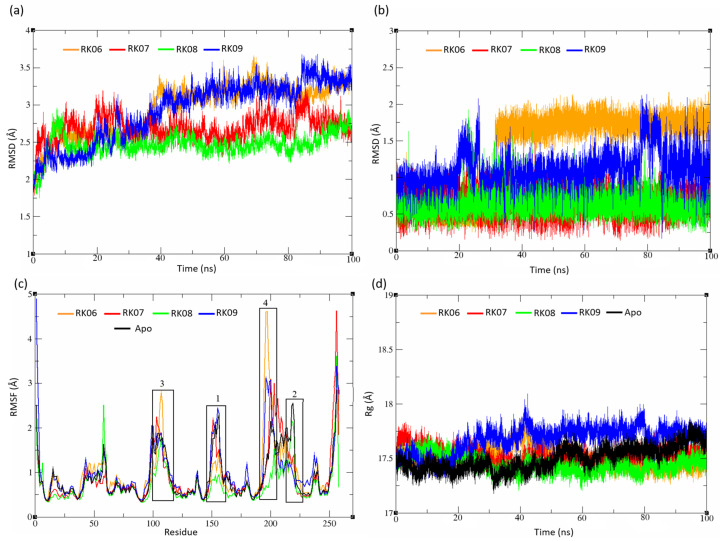
Substituted Diphenylmethane set analysis. (**a**) Complex RMSD, (**b**) ligand RMSD, (**c**) backbone RMSF and (**d**) the radius of gyration. RK06 and RK09 showed less stable systems compared to RK07 and RK08. All systems demonstrated almost the same protein compactness.

**Figure 7 molecules-28-03000-f007:**
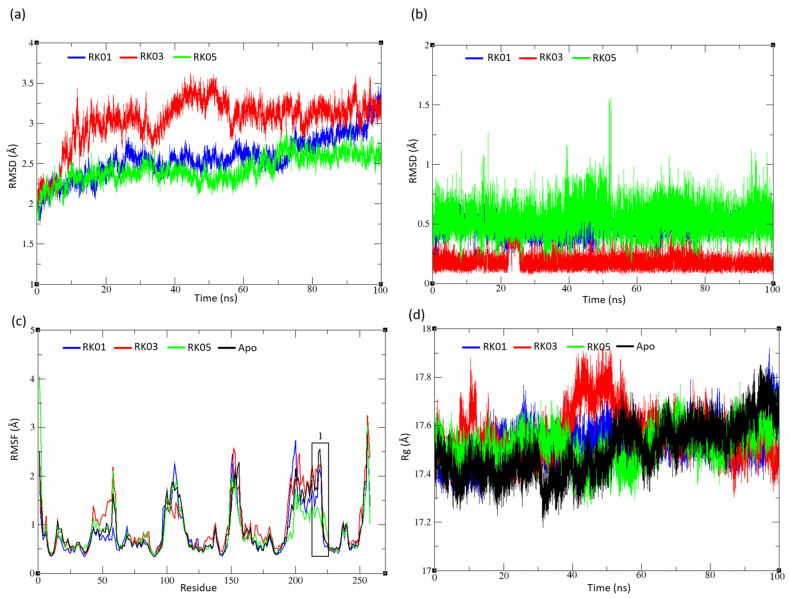
Diverse set analysis. (**a**) Complex RMSD, (**b**) ligand RMSD, (**c**) backbone RMSF, and (**d**) the radius of gyration. RK01 and RK05 reached the stable state after 10 ns of MD simulation compared to RK03, which showed an unstable system through the first 60 ns of MD simulation.

**Figure 8 molecules-28-03000-f008:**
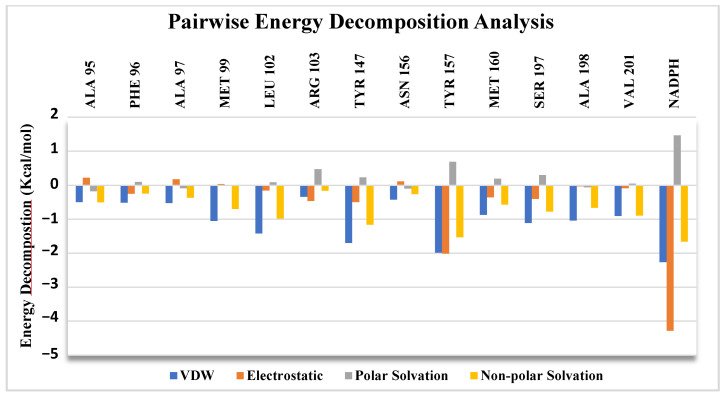
Pairwise energy decomposition for RK10. The highest contribution was seen with NADPH and the key residue Tyr 157.

**Figure 9 molecules-28-03000-f009:**
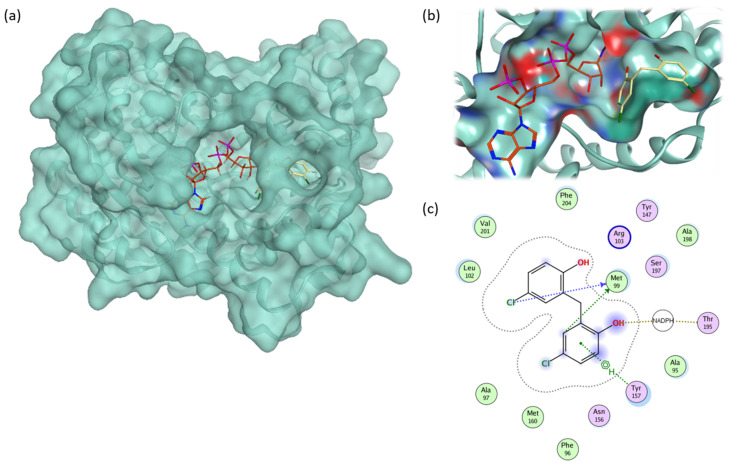
(**a**) Protein conformation for RK10 complex. (**b**) Fitting of the RK10 molecule (yellow stick) and NADPH (orange stick) inside the FabI active site. (**c**) The ligand interaction diagram for RK10. RK10 shows a good fitting inside the Fabi binding pocket where it interacts with Met 99, Tyr 157 and NADPH.

**Table 1 molecules-28-03000-t001:** The antimicrobial activity of ten compounds tested against *MRSA*, *E. coli*, *B. subtilis* and *S. aureus* using the disk diffusion test.

Compounds	Zone of Inhibition (mm)			
	*MRSA*	*E. coli*	*B. subtilis*	*S. aureus*
RK01	++	-	++	+
RK02	++	++	+++	++
RK03	+++	++	+++	+++
RK04	+++	-	++++	+++
RK05	++	-	-	+
RK06	-	-	-	-
RK07	-	-	-	-
RK08	++++	-	+++	+++
RK09	++++	-	++++	++++
RK10	++++	++	++++	++++
Chloramphenicol	+	++++	++++	+++
TCL ^†^	++++	+++	++++	++++
Dimethyl Sulfoxide	-	-	-	-

Key to symbols:—inactive (inhibition zone ≤6 mm): slight activity = + (inhibition zone >6–≤9 mm): moderate activity = ++ (inhibition zone >9–≤13 mm): high activity = +++ (inhibition zone >13–≤18 mm): very high activity = ++++ (>18). ^†^ Zone of inhibition values of TCL obtained from the literature.

**Table 2 molecules-28-03000-t002:** The MIC values (μg/mL) of our best hits along with the positive standard tested against different bacterial stains.

ID	MIC (μg/mL)			
	*MRSA*	*E. coli*	*B. subtillis*	*S. aureus*
MN02	8.00	32.00	4.00	4.00
**TCL**	64.00 ^†^	0.50 ^†^	0.40 ^†^	0.13 ^†^
Chloramphenicol	74.00 ^†^	4.00 ^†^	4.00 ^†^	4.00 ^†^
RK01	13.39	-	13.39	27.45
RK02	41.54	-	41.54	41.54
RK03	2.29	-	9.64	9.64
RK04	20.10	-	9.81	9.81
RK05	22.34	-	-	10.90
RK06	-	-	-	-
RK07	-	-	-	-
RK08	15.67	-	7.64	15.67
RK09	2.84	-	1.39	2.84
RK10	2.70	47.64	1.32	1.32

^†^ MIC values obtained from the literature.

**Table 3 molecules-28-03000-t003:** The docking and MM-GBSA scores along with the MIC values of the tested set of MN02 derivatives.

Molecule ID	Docking Score (kcal/mol)	MM-GBSA (kcal/mol)	MIC (μg/mL) *S. aureus*
MN02	−7.94	−34.60	4.00
TCL	−8.70	−32.32	0.13 ^†^
RK01	−6.03	−27.22	27.45
RK02	−9.18	−25.97	41.54
RK03	−8.71	−23.47	9.64
RK04	−8.87	−29.71	9.81
RK05	−7.48	−28.08	10.90
RK06	−8.33	−24.95	-
RK07	−9.21	−22.62	-
RK08	−8.53	−29.63	15.67
RK09	−8.88	−30.26	2.84
RK10	−8.00	−32.92	1.32

^†^ MIC values obtained from the literature.

## Data Availability

The authors confirm that the data supporting the findings of this study are available within the article and its Appendix A. Additional data are available on reasonable request from the corresponding author.

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
