# Peer review of "Molecular Modelling Study and Antibacterial Evaluation of Diphenylmethane Derivatives as Potential FabI Inhibitors"

_molecules, 2023, doi:10.3390/molecules28073000_

Round 1

Reviewer 1 Report

The paper by Hasan et al. aims at the identification of small inhibitors of bacterial enoyl‐acyl carrier protein reductase (FabI) An enzyme playing an important role in the biosynthesis of lipids and fatty acids essential for the integrity of the bacterial cell membrane and therefore, a target for the development of new antibiotics. The study is based on a) the use of a previously identified compound MN02 to carry out a similarity search in ligand databases., b) the selection of 10 compounds, their computational assessment and antimicrobial testing and finally c) the identification of RK10 which is claimed to be a promising candidate for developing broad-spectrum antibacterial agents.

The results described in the paper are interesting and the information provided can be useful for many researchers since the need for new antibiotics is an urgent global challenge due to the rapid emergence of bacterial strains resistant to existing drugs.

However, the description of these results in the manuscript needs major improvement before publication

Points to improve before publication:

One of the main limitations of this study is the lack of direct evidence that the observed antibacterial activity of the phenylmethane derivatives is due to the binding and inhibition of the proposed target FabI. In my opinion further research would be needed to confirm that the identified compounds are specific for FabI and to support the claims made in the manuscript. I recommend that a more detailed discussion to support that the identified compounds are specific for FabI is included in the manuscript.

Another potential limitation of the study is the number of bacterial species tested (MRSA, E. coli, B. subtilis and S. aureus). The use of the antibacterial activities of these compounds is useful to select the best hits, but a more extensive study of the antibacterial activities of the selected compounds would be needed to draw reliable conclusions on their potential use as antibiotics. This fact should be indicated and discussed in the text. In the same way authors claim that RK10 acts both on Gram-negative and Gram-positive bacteria while only E. coli has been tested as Gram-negative.

The manuscript should be carefully reviewed in detail and improved. Information could be organized in a better way. For instance:

Figure 1. I would recommend modifying Figure 1 for clarity. Taking care that all reactions are complete (i.e. FabH reaction is incomplete), and separating the second and following rounds of the FA biosynthetic pathway, from the last series of reactions which yield palmitoyl-ACP.

Moreover, Figure Legend should be self-explanatory to the reader and include all necessary information to understand it without reading the rest of the manuscript (The same applies to Figures 2 to 7)

Table 3. I would recommend removing this table or include it as supplementary information. Most of the results summarized in the table could be easily explained in the text.

Minor points:

Avoid using in the text terms that are too informal or casual. For instance:

             Line 71 “forming the “famous” hydrogen bonding….

             Line 132 one should “keep an eye on” ….

             Line 405 closely related in structure to the “famous” FabI inhibitor triclosan

References

             Use the same format for all references. In some of them DOI is missing

             Some references are incomplete: 50, 52, 57…

             Reference 66. Indicate when it was accessed…

Abbreviations should be defined the first time they appear in the text, even if they seem obvious to the authors

             For instance, MRSA is described in line 293 (!)

             GDP (line 40) is not explained

I would recommend not to include abbreviations in the abstract

Reviewer 2 Report

In This manuscript titled Molecular Modelling study and antibacterial evaluation of di-phenylmethane derivatives as potential FabI inhibitors, ten compounds were chosen for computational assessment and antimicrobial testing. The result shows that these compounds showed antibacterial activity against Gram-positive strains, and one of the compounds, RK10, exhibits broad-spectrum activity against both Grampositive and Gram-negative bacteria. The docking study shows that RK10 was able to exhibit the best fitting with the target catalytic pocket. the author verified the hypotheses and conclusions by the presented data, but several questions still need to be answered.

First, The TCL here is a positive control, or does the compound needs to be screened? It shows in some tables but not in others.

Second, do you run any ligand binding assay for these compounds with FabI to prove that these di-phenylmethane derivatives are the FabI inhibitors or these compounds may go through another mechanism to kill the bacteria? 

Third, These compounds show better inhibition with gram-positive bacteria compared with gram-negative bacteria, What the reason do you think?

Forth, What reason do you think that the MM-GBSA score is more fitting for the experiment?

Fifth, For different gram-positive bacteria, the docking score could not fit well, For example, RKO3 with -23.47kcal/mol for MM-GBSA, but show 2.29 MIC for MRSA. Do this mean it may have another unknown mechanism for these compound to go through and kill the bacteria? So some of the compounds may be not  FabI inhibitors?

Round 2

Reviewer 2 Report

After revision, The manuscript looks better and I also suggest checking the grammar and spelling.